# High-Efficiency 3D-Printed Three-Chamber Electromagnetic Peristaltic Micropump

**DOI:** 10.3390/mi14020257

**Published:** 2023-01-19

**Authors:** He Chen, Xiaodan Miao, Hongguang Lu, Shihai Liu, Zhuoqing Yang

**Affiliations:** 1School of Mechanical and Automotive Engineering, Shanghai University of Engineering Science, Shanghai 201620, China; 2National Key Laboratory of Science and Technology on Micro/Nano Fabrication, Shanghai Jiao Tong University, Shanghai 200240, China

**Keywords:** electromagnetic, peristaltic micropump, cantilever valve, 3D printing

## Abstract

This paper describes the design and characteristics of a three-chamber electromagnetic-driven peristaltic micropump based on 3D-printing technology. The micropump is composed of an NdFeB permanent magnet, a polydimethylsiloxane (PDMS) film, a 3D-printing pump body, bolts, electromagnets and a cantilever valve. Through simulation analysis and experiments using a single chamber and three chambers, valved and valveless, as well as different starting modes, the results were optimized. Finally, it is concluded that the performance of the three-chamber valved model is optimal under synchronous starting conditions. The measurement results show that the maximum output flow and back pressure of the 5 V, 0.3 A drive source are 2407.2 μL/min and 1127 Pa, respectively. The maximum specific flow and back pressure of the micropump system are 534.9 μL/min∙W and 250.4 Pa/W, respectively.

## 1. Introduction

In recent years, microfluidic system technologies have made significant progress [1]. Basic microfluidic components such as microchannels, microvalves, micropumps, microreactors, and various sensors and test platforms have been successfully integrated into the field of microfluidics and micro-electro-mechanical systems (MEMS). As the core component of the microfluidic system, the micropump provides the necessary power for the movement of the microfluidics. At the same time, the size and performance of the micropump has influence on many research fields such as biology, medicine, chemistry, and engineering [2,3]. The working principle of the micropump is to change the internal volume of the chamber through different driving forces to suction or discharge fluid, or directly promote the fluid flow in the pipeline. According to different driving forces, micropumps can be divided into mechanical micropumps (with moving parts) and non-mechanical micropumps (without moving parts) [4]. The mechanical micropump consists of a reciprocating piston or a deformable diaphragm, which apply pressure on the fluid by changing the volume of the pump chamber. As a result, the pumping liquid is realized. Reciprocating membrane micropumps are mainly driven by piezoelectric, electrostatic, electromagnetic, and other driving forms. Piezoelectric actuator micropumps have been widely researched and developed. Yun Hao Peng et al. developed a multi-channel piezoelectric micropump [5]. Through theoretical analysis, simulation, and experiments, systematic research on the combination of different driving units has been carried out. Even though piezoelectric actuator has certain advantages in response speed, controllability, and technology, its defects of high voltage, small deflection, and serious thermal radiation still exist [6]. However, electromagnetic actuators can provide driving force at low voltage, and Huawei Yu et al. proposed a magnetically driven micropump [7]. An electromagnetic actuator provides several paramount benefits for a micropump which have a low actuating voltage, fast response time, high field energy density, and wider operational range [8]. However, it has the disadvantage of a large volume and a complicated manufacturing process. Non-mechanical micropumps mainly rely on electrochemical performance to push charged fluid through electrostatic force for movement [9]. Peristalsis is the basic transport mechanism of living organs and organisms. It can realize the functions of the digestive tract, urinary tract transport, the blood flow of the circulatory system, and so on. Peristaltic micropumps usually have three or more chambers, which operate in a specific sequence to deliver fluid to a desired direction. At present, most of the peristaltic micropumps are driven by piezoelectricity. Aiming at the defects of high voltage and small deflection of piezoelectric drive, electromagnetic drive has the advantages of a low voltage, wider working range, high energy density, etc. Therefore, in this paper, electromagnetic drive is proposed for liquid pumping to solve the problem of high voltage required by piezoelectric drive.

In this paper, a three-chamber peristaltic micropump with a simple structure is designed. The permanent magnet on the membrane is driven by an external electromagnetic field. The purpose of pumping liquid is achieved by changing the volume of the chamber. The single-chamber, multi-chamber, valved, and valveless cases are discussed through simulation analysis.

## 2. Principle and Design

### 2.1. Physical Model

In order to evaluate the performance of the electromagnetic-driven micropump, the prototype pump shown in Figure 1 was manufactured using 3D-printing technology. The model was manufactured by stereo lithography apparatus (SLA) 3D-printing technology. The size of the single-chamber micropump is 35 mm × 35 mm × 8 mm, and it is made of translucent resin. The electromagnetic micropump is mainly composed of an NdFeB permanent magnet, a PDMS film, a pump body, a one-way check valve, a bolt, and an electromagnet. The one-way check valve is set at the inner inlet side of the pump chamber. The diameter of the pump chamber is 26 mm, and the depth is 6 mm. PDMS film plays a sealing role in the pump chamber, and acts as a driving film to change the volume of the chamber. The electromagnet is placed at the bottom of the pump body, and the bolt plays the role of sealing the pump body. The size of the three-chamber micropump is 100 mm × 35 mm × 8 mm, and the single chamber series connection is adopted.

The working process of the micropump includes three different states (Figure 2). In the equilibrium state, the pump membrane is in the equilibrium position; that is, no pressure is generated on the pump chamber. In the suction state, the repulsive electromagnetic driving force is generated on the pump membrane, driving the membrane upward. The pressure in the pump chamber is negative and the external fluid is sucked into the pump chamber from the left inlet. In the pumping state, the permanent magnet on the pump membrane is attracted by the electromagnetic driving force, driving the pump membrane downward, thus squeezing the fluid in the chamber and pumping it out from the right outlet. By changing the current direction of the electromagnetic driver, the suction and pumping working states could be realized repeatedly.

### 2.2. Electromagnetic Theory

In this study, the electromagnetic force is used to drive the micropump. When the electromagnet is energized, it attracts or repels the permanent magnet on the PDMS film to change the volume of the pump chamber, so as to suck and pump liquid. The electromagnet made in the factory was used in this study. According to the empirical formula:(1)$$F=\frac{\emptyset^{2}}{2\mu_{0}S}=\frac{B^{2}}{2\mu_{0}}S$$
where ∅ is the working air gap magnetic flux, Wb; *B* is the magnetic induction strength of the working air gap, T; *μ*_0_ is the vacuum permeability and its value is 4π × 10^−7^ Wb/A∙m; *S* is the cross-sectional area of magnetic circuit, m^2^. Ideally, regardless of the air gap, the magnetic induction strength *B* is:(2)$$B=\frac{N\cdot U}{R\delta }\mu_{0}=\frac{NI}{\delta }\mu_{0}$$
where *N* is the number of coil turns; *I* is current intensity, A; *U* is the power supply voltage, V; *R* is winding resistance, Ω; *δ* is the air gap length, m. 

Substituting Formula (2) into Formula (1), we can obtain:(3)$$F=\frac{{\left(NI\right)}^{2}\mu_{0}}{2\delta^{2}}S$$

It can be seen that the magnitude of the electromagnetic force is directly proportional to the square of the product of the number of turns of the coil and the magnitude of the current.

### 2.3. PDMS Membrane Deformation Theory

At present, a large number of PDMS applications have been successfully realized in the literature; however, the deformation behavior of PDMS films is still unclear. Therefore, this paper theoretically analyzes the deformation variable of films through the Timoshenko model. In this model, it is assumed that $w^{*}\left(r^{*}\right)$ is the same as in the literature, but $u^{*}\left(r^{*}\right)$ is different [10,11]. There is therefore the following formula:(4)$$U_{T}=\frac{8\pi Eh^{3}}{9\left(1-\nu^{2}\right)}\frac{w_{0}^{2}}{a^{2}}+\frac{2\pi Eh}{189}\frac{\left(19-5\nu \right)}{\left(1-\nu \right)}\frac{w_{0}^{4}}{a^{2}}-\frac{\pi Pw_{0}a^{2}}{3}$$

For the deformation of the film, the total energy ($U_{T}$) consists of strain energy, tensile energy, and external pressure, where E (Young’s modulus), h (PDMS membrane thickness), ν (Poisson’s ratio), $w_{0}$ (maximum membrane deformation), a (membrane radius), and P (external pressure).

When the membrane reaches the maximum deformation, the derivative of the total energy approaches zero ($U_{T}∕dw_{0}=0$). Therefore, the value of $w_{0}/h$ can be expressed as:(5)$$\frac{16}{3\left(1-\nu^{2}\right)}\left(\frac{w_{0}}{h}\right)+\frac{8\left(19-5\nu \right)}{63\left(1-\nu \right)}{\left(\frac{w_{0}}{h}\right)}^{3}=\frac{Pa^{4}}{Eh^{4}}$$

Substitute Poisson’s ratio 0.4 into Equation (5) to obtain:(6)$$\frac{400}{63}\left(\frac{w_{0}}{h}\right)+\frac{680}{189}{\left(\frac{w_{0}}{h}\right)}^{3}=\frac{Pa^{4}}{Eh^{4}}$$

The above formula is used to predict the maximum deformation of the membrane under the parameters of the membrane radius, thickness, Young’s modulus, and driving air pressure. 

Considering the maximum deflection of the membrane, there are $\omega_{0}∕h\gg 1$. The above formula can be simplified as:(7)$$w_{0}=0.65a{\left(\frac{Pa}{Eh}\right)}^{\frac{1}{3}}$$

## 3. Finite Element Simulation

The working principle of the electromagnetic micropump involved the electric field, the magnetic field, the fluidic flow dynamics, and the elastic deformation of the PDMS film, which increased the complexity and difficulty of simulation. In order to visualize the output performance of the micropump, the electromagnetic force calculated by the simulation is converted to the pressure applied on the film. 

Fluent 2020R2 was used to simulate and analyze the single chamber without the valve and the single chamber with the valve. In the valveless simulation, the inlet and outlet are defined as open states. On the contrary, in the valved state, the inlet is defined as closed and the outlet is open.

First, the single-chamber model was simulated and analyzed. In Figure 3, the output back pressure of the single chamber with valves is shown to be better than that of the single chamber without valves. Under the steady state, the output back pressure of the single chamber with the valve can reach 289 Pa, and the back pressure of the single chamber without the valve is 216 Pa. As the simulation results indicate, it can be concluded that the check valve plays a role of pressure leakage in the process of pumping liquid, and can improve the output back pressure of the single-chamber model.

This paper adopts a three-chamber series model, which includes two different starting modes. The first startup mode is synchronous startup (Figure 4a), which means that the three chambers enter the pumping state synchronously. At this time, the fluid inside the pump chamber is pumped out of the chamber from the right outlet. The second startup mode is sequential startup (Figure 4b), in which three chambers enter the pumping state from the first chamber to the third chamber in sequence, and the chambers remain in the pumping state until the third chamber enters the pumping state. At this time, the fluid is pushed from the first chamber to the third chamber through traveling waves and then pumped out from the outlet. Figure 5a shows the simulation results of two startup modes of the multi-chamber valveless system. As shown in Figure 5a, three-chamber valveless system output can reach 480 Pa pressure under the synchronous startup condition, and only 322 Pa under the sequential startup condition. Figure 5b shows the simulation results of two startup modes with multi-chamber valves. In the case of valves, the sequential startup mode output can reach 1121 Pa pressure, while the synchronous startup mode output can only reach 243 Pa pressure. Figure 5c shows the comparison of working conditions of sequential startup in the valved and valveless models. It can be concluded that the sequential startup with valves is better than the sequential startup without valves.

According to the analysis results, in the valveless model, since there is no check valve, the fluid between the chambers can flow into each other, and the pressure of the three chambers output can be realized at the same time under the synchronous start-up condition. However, under the sequential start-up condition, since there is no check function of the check valve, the start-up of the latter chamber will cause some liquid flow back to the previous chamber, resulting in pressure leakage. In the valved model, the sequential start-up condition is obviously better than the synchronous start-up condition. The main reason for this phenomenon is that the check valve hinders the mutual transmission of pressure between chambers during synchronous start-up, thus affecting the final output back pressure. To sum up, the multi-chamber and valved model is selected in the model, and the sequential starting condition is selected in the starting condition, which is the best scheme according to the simulation results. This simulation can provide a scheme reference for subsequent tests.

## 4. Measurement and Evaluation

The micropump was manufactured using 3D-printing technology, using an Arduino development board as the control circuit and Keysight B2901A as the output power supply. Three electromagnets are used as driving magnets (Table 1) for specific parameters, and the NdFeB permanent magnets are used as on-film magnetic poles (Figure 6). The liquid level in the liquid outlet pipe before pumping is compared with the liquid level after pumping by taking pictures with a high-speed camera. The parameter performance of the micropump can be obtained by measuring the liquid level in the liquid outlet pipe through calculation.

In the single-chamber test, in order to test the pumping performance of the micropump, water is first injected into the chamber to ensure that there is no air gap in the pump chamber after water injection. The purpose is to achieve greater pumping pressure in the pumping process. The water injection height should be flushed with the water surface of the micropump chamber to ensure the accuracy of the experimental results. The performance of the single chamber indirectly reflects the performance of three chambers. The test is divided into valveless (Figure 7a) and valved (Figure 7b). Three chambers are connected in series, and the chambers are numbered from left to right. The left chamber is chamber 1, the middle chamber is chamber 2, and the right chamber is chamber 3. Three chambers can be divided into a valveless state (Figure 7c) and a valved state (Figure 7d).

## 5. Results and Discussion

Figure 8a describes the relationship between the pumping pressure and the number of permanent magnets in the single-chamber valveless and valved states, respectively. It can be seen from Figure 8a that the output pressure of the valved model increases with the increase in the number of magnetic blocks. When there are five magnets, the output back pressure can reach 441 Pa, which is 925% higher than that of the valveless model. At this time, the valveless model can only output a pressure of 43 Pa. The reason is that when the number of magnetic blocks exceeds three. The PDMS diaphragm starts to compress downward under the action of the magnetic block’s gravity, making it emerge out of equilibrium and into pumping state. The liquid in the chamber is pushed out of the chamber, resulting in there being not enough liquid in the chamber at startup, and the corresponding pressure cannot be pumped out. In the valved model, due to the existence of the check valve, the liquid in the chamber will not flow out from the inlet, and even if the number of magnetic blocks reaches as many as five, the micropump can always be kept at the equilibrium position before starting, but at this time, due to the excessive number of magnetic blocks, the driving force of the micropump is mainly provided by the gravity of the magnetic block.

Figure 8b is a three-chamber valveless model, including a synchronous startup mode and sequential startup mode. Table 2 corresponds to the number of magnetic blocks driving each chamber. In the valveless model, when the number of magnetic blocks is three, four, and five, respectively, the output pressure can reach 490 Pa, and the sequential start can only reach 401 Pa. Compared with the sequential start, the pumping performance of the synchronous start decreases by 7%. Figure 8c shows the comparison of output pressure between the synchronous start-up and the sequential start-up of the multi-chamber valved model. When the number of drive magnetic blocks is three, four, and five, the output back pressure of the synchronous start-up can reach 431.2 Pa, and the back pressure of the sequential start-up can reach 1127 Pa. The performance of the sequential start-up is 161% higher than that of the synchronous start-up. Figure 8d is a comparison of the multi-chamber valveless and valved models under sequential starting conditions. It can be concluded that in the multi-chamber micropump, sequential starting with valves is the best working mode.

## 6. Conclusions

In this paper, a peristaltic micropump model based on electromagnetic drive is proposed. The model was manufactured through SLA 3D-printing technology. PDMS film is used as its driving film. The permanent magnet fixed on the PDMS film is driven by the alternating magnetic field generated by the alternating current input to the electromagnet to change the chamber volume, so as to achieve the purpose of pumping fluid. Through simulation analysis and experimental testing, the following conclusions are drawn.

The simulation and experimental data show that the addition of a one-way check valve is helpful to improve the output back pressure. In the single-chamber valved model, 289.7 Pa output back pressure can be obtained from simulation data, and 254.8 Pa output back pressure can be obtained from experimentation. The error between the experiment and simulation data was found to be 13.7%. In the single-cavity model, the output back pressure can be increased by increasing the number of permanent magnets.

In the three-chamber valved model, the output back pressure of 1127 Pa can be obtained through the sequential startup, and the number of permanent magnets in each chamber is three, four, and five in turn. This is because in the process of sequential starting, the pressure of the current chamber is transmitted to the next chamber after starting, and the initial pressure of the latter chamber becomes larger when starting, so that more driving force is needed to drive and more permanent magnets are needed to provide more driving force. The simulation results show that the output back pressure is 1121.6 Pa, the experimental results are 1127 Pa, and the actual error is only 0.5%.

Therefore, an electromagnetic-driven three-chamber peristaltic micropump is constructed in this research. The driving voltage is only 5 V, the driving current is only 0.3 A, and the total power of the three electromagnetic actuators is 4.5 W. Under the sequential startup mode, the output back pressure of 1127 Pa can be provided, and the flow can reach 2407.2 μL/min. Table 3 summarizes the characteristics of our micropumps and previously developed electromagnetic micropumps reported in other articles in the literature. The maximum specific flow and back pressure can be 534.9 μL/min·W and 250.4 Pa/W, respectively, which shows that the model has the performance of low voltage, low power, and high output, and provides two feasible schemes for the subsequent electromagnetic micropump.

In contrast to the electromagnetic micropumps reported in other articles, the 3D-printing process is adopted in this research, which is simple in the manufacturing process, low in cost, and reliable in performance. Although the model has a large volume, its output performance has significant advantages over those of other micropumps. The electromagnetic-driven peristaltic pump reported in this paper can be used for insulin supply. In the future, MEMS can be used for processing. The use of magnetic PDMS film and flat coil can further reduce the size of the micropump.

## Figures and Tables

**Figure 1 micromachines-14-00257-f001:**
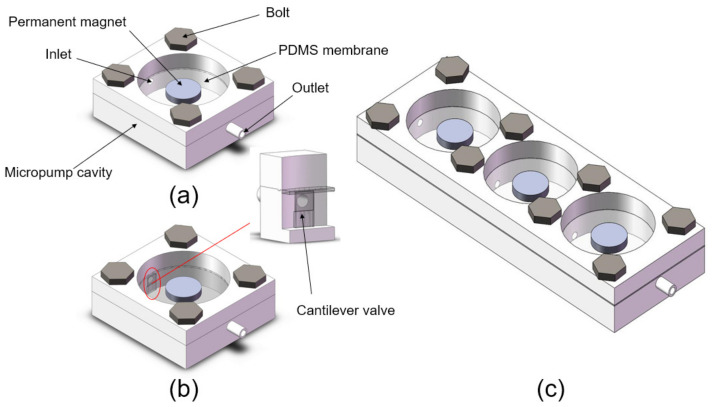
(**a**) Single-chamber prototype pump without a valve; (**b**) single-chamber prototype pump with valve; (**c**) three-chamber prototype pump.

**Figure 2 micromachines-14-00257-f002:**
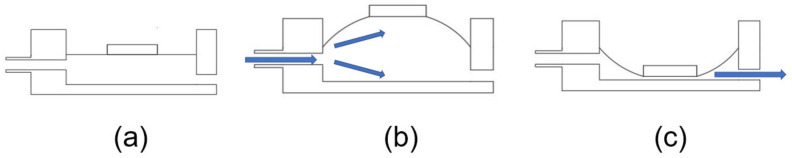
(**a**) The equilibrium state; (**b**) the suction state; (**c**) the pumping state.

**Figure 3 micromachines-14-00257-f003:**
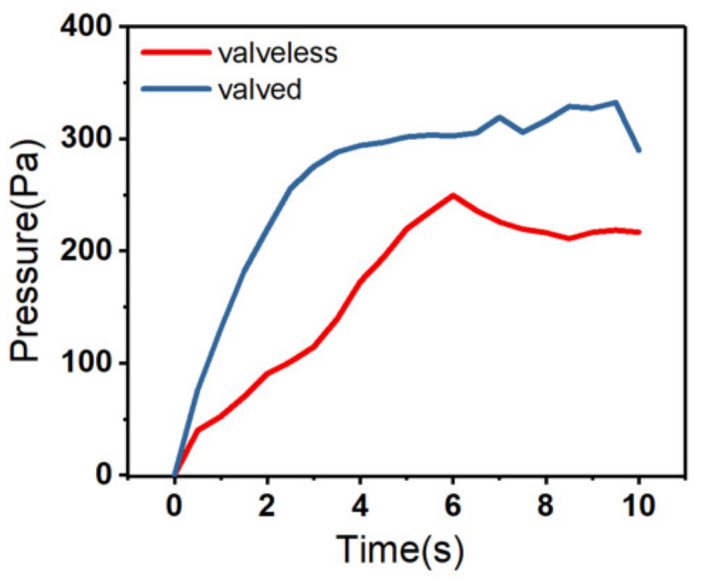
Single-chamber valveless and valved output pressure.

**Figure 4 micromachines-14-00257-f004:**
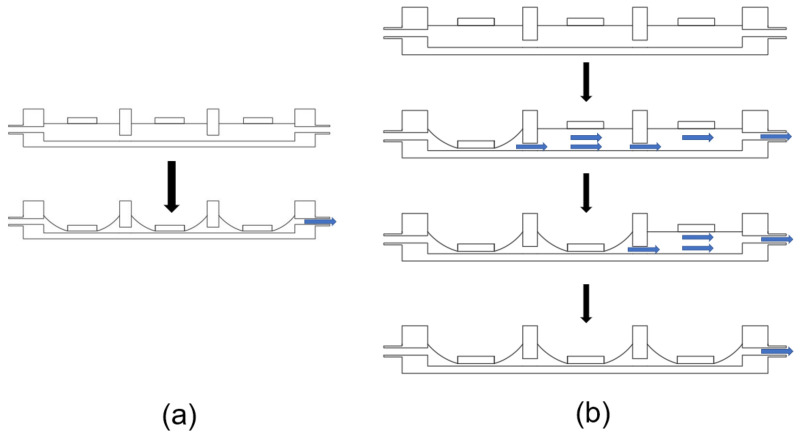
Startup mode: (**a**) synchronous start mode; (**b**) sequential start mode.

**Figure 5 micromachines-14-00257-f005:**
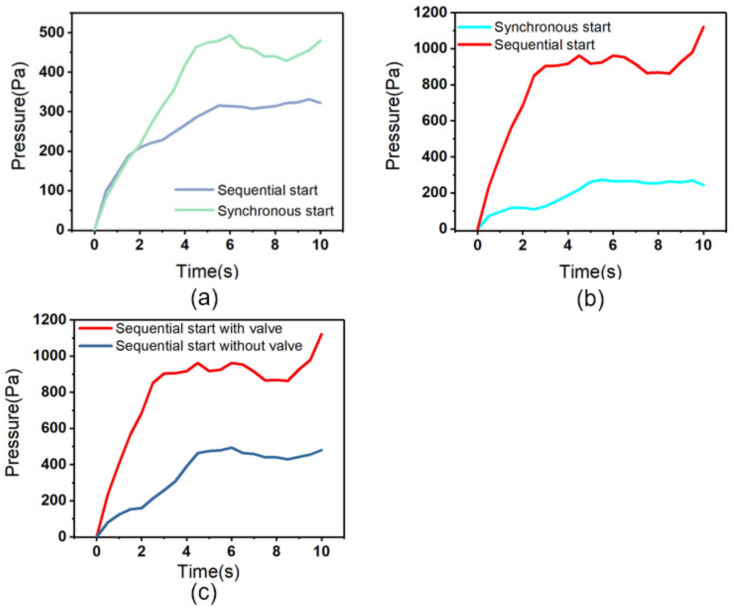
Three-chamber simulation output pressure: (**a**) three-chamber valveless system; (**b**) three-chamber with valve system; (**c**) sequential startup in the valved and valveless models.

**Figure 6 micromachines-14-00257-f006:**
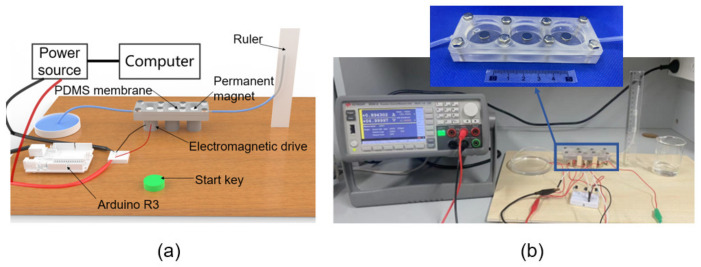
Experimental platform: (**a**) rendering of experimental platform, (**b**) experimental platform (the enlarged view is fabricated device).

**Figure 7 micromachines-14-00257-f007:**
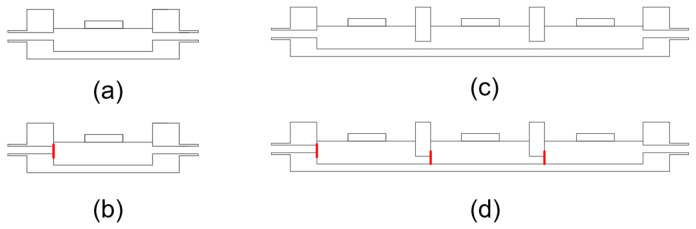
Single-chamber model: (**a**) valveless model; (**b**) valved model. Three-chamber model: (**c**) valveless model; (**d**) valved model.

**Figure 8 micromachines-14-00257-f008:**
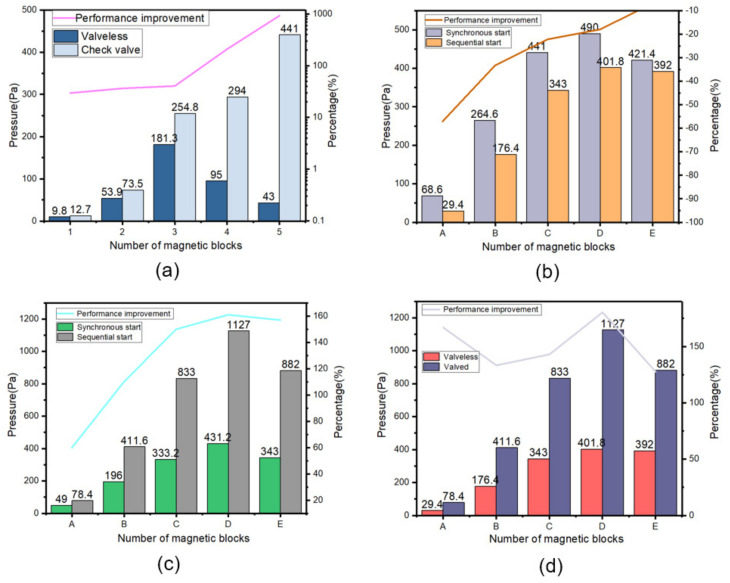
(**a**) Single-chamber valveless and valved output pressure, (**b**) three-chamber model without valve, (**c**) three-chamber model with valve, (**d**) sequential start mode.

**Table 1 micromachines-14-00257-t001:** Electromagnet parameters.

Electromagnet Parameters	
Rated current	0.3 A
Rated voltage	5 V
Rated power	1.5 W
Size	20 mm × 20 mm

**Table 2 micromachines-14-00257-t002:** Number of magnetic blocks in each chamber.

Number of Magnetic Blocks	
A	Each chamber has two magnetic blocks
B	Each chamber has three magnetic blocks
C	There are three, four, four and a half magnetic blocks in turn
D	There are three, four, and five magnetic blocks in turn
E	There are three, four, five and a half magnetic blocks in turn

**Table 3 micromachines-14-00257-t003:** Performance comparison of electromagnetic micropumps.

Ref.	Q_max_	P_max_	Voltage	Current	Power	Specific Flow	Specific Back Pressure
Unit	μL/min	Pa	V	A	W	μL/min·W	Pa/W
Y.H. [12]	0.17	N/A	20	0.12	2.4	0.071	N/A
M.T. [13]	1.25	N/A	N/A	N/A	2.2	0.568	N/A
R.G. [14]	51	350	7.5	0.13	0.975	52.308	358.97
X.M. [15]	8915.8	356.72	9	2.5	22.5	396.26	15.85
This work	2407.2	1127	5	0.9	4.5	534.933	250.4

## Data Availability

Not applicable.

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
