# Peer review of "High-Efficiency 3D-Printed Three-Chamber Electromagnetic Peristaltic Micropump"

_micromachines, 2023, doi:10.3390/mi14020257_

Round 1

Reviewer 1 Report

The authors presented an electromagnetic peristaltic micropump using permanent magnets, 3D printed chambers, and flexible PDMS film. The work is comprehensive with including simulation analysis and experimental testing and validation. 

Kindly note the following comments:

- The manuscript doesn't seem to be fitting the scope of the journal special issue "Flexible and Wearable Sensors". Considering other venues is highly recommended. 

- The concept of linear peristaltic pumps driven by magnetic actuators have been reported widely in the last decade with different level of efforts such as performance improvement or device miniaturization. The engineering work in this manuscript is outstanding, but the novelty aspect is not clear. Also, the targeted application for this technology is not clarified.

- The overview of the peristaltic micropump in the manuscript is limited and doesn't highlight the breakthrough in this field (e.g. miniaturization, precision, using innovative actuation materials, ..etc). Also, the challenges need to be discussed thoroughly and aligned with the motivation of this work. For example: The high voltage issue mentioned as a challenge for PZT actuators has already been addressed by other peristaltic micropump technologies that utilize alternative actuation techniques leading to low voltage/power systems. 

-Further analysis on the flow rate control and consistency during extended operation is suggested. The pressure analysis is a useful measure but several review papers study the synchronization patterns and its influence on the flow rate. This would be useful to investigate for this actuation method to control flow rate and minimize power consumption. Also, some hysteresis is expected from the continuous attraction/repelling forces (intrinsic magnetic property of the magnet or field variation of the electromagnet), as well as the hysteresis relevant to the elasticity of the membrane. These expected variations will affect the flow rate consistency in the extended operation test.

Author Response

We are grateful to the reviewer for the constructive suggestions, and the response and revisions are as follows:

Comment 1. The manuscript doesn't seem to be fitting the scope of the journal special issue "Flexible and Wearable Sensors". Considering other venues is highly recommended.

Comment 2. The concept of linear peristaltic pumps driven by magnetic actuators have been reported widely in the last decade with different level of efforts such as performance improvement or device miniaturization. The engineering work in this manuscript is outstanding, but the novelty aspect is not clear. Also, the targeted application for this technology is not clarified.

Response: We are grateful to the reviewer for the constructive suggestion. In the past decade, most magnetic actuators are composed of a motor and a magnetic transmission. and there are few electromagnetic drives. The reason why the electromagnetic drive is selected in this paper is to make a feasibility study for further research. This device can be used for insulin supply. In the future, MEMS technology will be used for micro pump processing. The magnetic film and flat coil design can greatly reduce its volume.

Comment 3. The overview of the peristaltic micropump in the manuscript is limited and doesn't highlight the breakthrough in this field (e.g. miniaturization, precision, using innovative actuation materials, ..etc). Also, the challenges need to be discussed thoroughly and aligned with the motivation of this work. For example: The high voltage issue mentioned as a challenge for PZT actuators has already been addressed by other peristaltic micropump technologies that utilize alternative actuation techniques leading to low voltage/power systems.

Response: We thank the reviewer for pointing out this issue, The table shows the advantages and disadvantages of each actuation method. It is obvious that the electromagnetic actuation method can not only reduce the drive voltage and current, but also has the advantages of simple structure and fast response speed

Actuation method

Advantages

Disadvantages

Piezoelectric

Large actuation force, Simple structure, Fast response

High operating voltage

Small stroke

Electrostatic

Fast response, Low power consumption

High operating voltage, Small stroke

Pneumatic

Simple-to-control actuation force, Large stroke

Slow response, High actuation pressure

Thermal(Shape memory alloy)

Low operating voltage, Simple driving circuit, Large stroke

High power consumption, Slow response

Motor

Low operating voltage, Large stroke

Complex external mechanism required for handling rotary

motion

Comment 4. Further analysis on the flow rate control and consistency during extended operation is suggested. The pressure analysis is a useful measure but several review papers study the synchronization patterns and its influence on the flow rate. This would be useful to investigate for this actuation method to control flow rate and minimize power consumption. Also, some hysteresis is expected from the continuous attraction/repelling forces (intrinsic magnetic property of the magnet or field variation of the electromagnet), as well as the hysteresis relevant to the elasticity of the membrane. These expected variations will affect the flow rate consistency in the extended operation test.

Response: Thanks for your kind advice. In fact, this paper is aimed at the research of insulin delivery pump. Although the output flow is an important parameter of the performance of the micropump, the output back pressure of the micropump is more worthy of attention in the research of micropump that inputs fluid to a pressurized environment. The test of output flow will be enhanced in subsequent studies.

Reviewer 2 Report

Manuscript ID: micromachines-2121544

REVIEWER REPORT:

Recommendation: Major revisions

Comments:

The paper "High efficiency 3D printed three-chamber electromagnetic peristaltic micropump" revealed a very similar report compared to the studies presented by the other groups in a few earlier publications.  

Major Comments:

1-      Authors have presented a peristaltic pump which, not even in theory, has valves in it. In general, a peristaltic pump is a positive displacement bidirectional flow valveless pump which displaces a volume of fluid from one point to another without the involvement of any valve structure. The sweeping action introduced by the roller-based assemblies in the peristaltic micropumps itself behaves like valves to restrict the backflow of the sample fluid. The author needs to comment upon the inclusion of valve structure, which is generally not found in a conventional peristaltic pump.

2-      Sequential activation of electromagnets placed under the PDMS membrane which adds delay in the volumetric displacement and will ultimately lead to the non-continuous fluid flow at the outlet end. On the other hand, there are many peristaltic micropumps which have more smooth flow profile. The author needs to comment on this.

3-      The currently proposed design seems more of 3 units of Diaphragm micropump combined together. Most of the Diaphragm micropumps have a unidirectional valve to ensure the unidirectional flow of the micropumps. Peristaltic micropumps are designed to pump in both forward and reverse directions which in the presented device is not possible because of the presence of the unidirectional check valves. The author needs to comment on this.

4-      The "conclusion" section of the manuscript needs to be restructured to highlight the key facts regarding the device plan, experimental progression, results, innovations, technological gains and achievements.

5-      In the conclusion or result section, the potential drawback of the proposed pumping technique needs to be also mentioned, along with future scope for further innovation in this concern.

6-      Recommend the potential amendment in the design and process (in the near future) to improve ahead of the current design (which has already been presented in an earlier report by the other group).

7-      Cost investigation, accessibility and merits/demerits of other related pumping techniques may be analysed in brief, as all techniques have pros and cons in terms of the maintenance cost, the flexibility of use in different ambiences, portability, installation cost, operation cost, manpower to operate, and medical safety. Comment upon all these points.

Minor Comments:

1-      Figures presented by the authors don’t provide a clear picture of the fabricated device.

2-      Positioning of the electromagnets is not incorporated in the device schematic.

3-      According to the Figure 2 & 4 schematic, the fluid net flow output will be zero as there the pumping chamber is open from both ends. Else mention in the caption the valveless flow condition.

4- The type of 3D printing technology is not mentioned in the manuscript.

5-    Are the valves also fabricated within the 3D design itself or fabricated separately and then incorporated? The author needs to comment on this because the fabrication of cantilever-based valves will affect the overall fabrication cost of the micropump.

Author Response

We are grateful to the reviewer for the valuable comments and suggestions, and the response and revision are as follows:

Comment 1. Authors have presented a peristaltic pump which, not even in theory, has valves in it. In general, a peristaltic pump is a positive displacement bidirectional flow valveless pump which displaces a volume of fluid from one point to another without the involvement of any valve structure. The sweeping action introduced by the roller-based assemblies in the peristaltic micropumps itself behaves like valves to restrict the backflow of the sample fluid. The author needs to comment upon the inclusion of valve structure, which is generally not found in a conventional peristaltic pump.

Response: We are grateful to the reviewer for the valuable comments and suggestions. In fact, in the valveless model, this model fully conforms to the definition of peristaltic pump. In the valved model, the introduction of valves is indeed inconsistent with the definition of peristaltic pumps. In this valved model, the forward pumping process of fluid still conforms to the definition of peristaltic pump, and the fluid is still pushed forward and pumped out by traveling waves. As the model is oriented to the market application of insulin delivery pump, valves are added in the model to ensure that no backflow occurs when the ambient pressure changes.

Comment 2. Sequential activation of electromagnets placed under the PDMS membrane which adds delay in the volumetric displacement and will ultimately lead to the non-continuous fluid flow at the outlet end. On the other hand, there are many peristaltic micropumps which have more smooth flow profile. The author needs to comment on this.

Response: We thank very much for this valuable comment. Electromagnetic drive has the characteristics of fast response, low power consumption and simple structure. The peristaltic micropump with smooth fluid profile often adopts other driving forms with high voltage, high input or more complex structure. The main research direction of this paper is to improve pumping performance on the premise of reducing drive power consumption. For the problems raised by the reviewer, this paper will focus on solving them in the subsequent research center.

Comment 3. The currently proposed design seems more of 3 units of Diaphragm micropump combined together. Most of the Diaphragm micropumps have a unidirectional valve to ensure the unidirectional flow of the micropumps. Peristaltic micropumps are designed to pump in both forward and reverse directions which in the presented device is not possible because of the presence of the unidirectional check valves. The author needs to comment on this.

Response: Thanks for your kind advice. In the valveless model, two-way flow can be carried out in strict accordance with the definition of peristaltic pump. As the model is oriented to the market application of insulin delivery pump, a valve is added to the model to ensure that no backflow occurs when the ambient pressure changes.

Comment 4. The "conclusion" section of the manuscript needs to be restructured to highlight the key facts regarding the device plan, experimental progression, results, innovations, technological gains and achievements.

Response: Thanks for your valuable suggestion. We have updated the conclusion (Page 8, line 256 to line 292).

Comment 5. In the conclusion or result section, the potential drawback of the proposed pumping technique needs to be also mentioned, along with future scope for further innovation in this concern.

Response: We thank the reviewer for pointing out this issue. We have added the issue in conclusion. (Page 8, line 288 to line 292).

Comment 6. Recommend the potential amendment in the design and process (in the near future) to improve ahead of the current design (which has already been presented in an earlier report by the other group).

Response: Thanks for the comment. We have added the issue in conclusion. (Page 8, line 288 to line 292).

Comment 7. Cost investigation, accessibility and merits/demerits of other related pumping techniques may be analysed in brief, as all techniques have pros and cons in terms of the maintenance cost, the flexibility of use in different ambiences, portability, installation cost, operation cost, manpower to operate, and medical safety. Comment upon all these points.

Response: We are grateful to the reviewer for the constructive suggestion. We have added new points in conclusion (Page 8, line 286 to line 288).

Minor Comments:

Comment 1. Figures presented by the authors don’t provide a clear picture of the fabricated device.

Response: Thanks for your kind advice. We have edited the Figure 6, added the enlarged view is fabricated device.

Comment 2. Positioning of the electromagnets is not incorporated in the device schematic.

Response: Thanks for your kind advice. The positioning of the electromagnet can be seen in Figure 6a. We have increased the font to clearly show the position of the electromagnet.

Comment 3. According to the Figure 2 & 4 schematic, the fluid net flow output will be zero as there the pumping chamber is open from both ends. Else mention in the caption the valveless flow condition.

Response: We are grateful to the reviewer for the valuable comments and suggestions. We redraw Figure 2 and Figure 4, explained the flow direction, and added text supplement (Page 3 line 86 to line 87, line 89 to line 90. Page 4 line 160 to line 161,line 164 to line 166).

Comment 4. The type of 3D printing technology is not mentioned in the manuscript.

Response: Thanks for the comment. The model is manufactured by SLA (Stereo Lithography Apparatus) 3D printing technology. We added the point in Page 2 line 68.

Comment 5. Are the valves also fabricated within the 3D design itself or fabricated separately and then incorporated? The author needs to comment on this because the fabrication of cantilever-based valves will affect the overall fabrication cost of the micropump.

Response: We are grateful to the reviewer for the constructive suggestion. The valves of this model are not manufactured by 3D printing process. It is cut through PET film and reassembled in the pump body.

Round 2

Reviewer 1 Report

I thank the authors for their response and for the modifications.

The manuscript is sufficiently improved and the comments are mostly addressed

Reviewer 2 Report

I have gone through the revised manuscript and according to my perspective, the manuscript has been improved and the authors addressed most of our comments, which is satisfactory.